# Development of the Home Cooking EnviRonment and Equipment Inventory Observation form (Home-CookERI^TM^): An Assessment of Content Validity, Face Validity, and Inter-Rater Agreement

**DOI:** 10.3390/nu12061853

**Published:** 2020-06-21

**Authors:** Sonja Schönberg, Roberta Asher, Samantha Stewart, Matthew J. Fenwick, Lee Ashton, Tamara Bucher, Klazine Van der Horst, Christopher Oldmeadow, Clare E. Collins, Vanessa A. Shrewsbury

**Affiliations:** 1Department Nutrition and Dietetics, Faculty of Health Professions, Bern University of Applied Sciences, Murtenstrasse 10, 3012 Bern, Switzerland; sonja.schoenberg@bfh.ch (S.S.); klazine.vanderhorst@bfh.ch (K.V.d.H.); 2Priority Research Centre for Physical Activity & Nutrition (PRCPAN), The University of Newcastle, 1 University Drive, Callaghan, NSW 2308, Australia; roberta.asher@newcastle.edu.au (R.A.); samantha.stewart@uon.edu.au (S.S.); matthew.j.fenwick@uon.edu.au (M.J.F.); lee.ashton@newcastle.edu.au (L.A.); tamara.bucher@newcastle.edu.au (T.B.); clare.collins@newcastle.edu.au (C.E.C.); 3School of Health Sciences, Faculty of Health and Medicine, The University of Newcastle, 1 University Drive, Callaghan, NSW 2308, Australia; 4School of Environmental and Life Sciences, Faculty of Science, The University of Newcastle, 10 Chittaway Road, Ourimbah, NSW 2258, Australia; 5School of Medicine and Public Health, Faculty of Health and Medicine, The University of Newcastle, 1 University Drive, Callaghan, NSW 2308, Australia; christopher.oldmeadow@newcastle.edu.au; 6Hunter Medical Research Institute (HMRI), New Lambton Heights, NSW 2305, Australia

**Keywords:** cooking environment, validity, inter-rater agreement, online survey, reproducibility

## Abstract

Introduction: Quantifying Home Cooking EnviRonments has applications in nutrition epidemiology, health promotion, and nutrition interventions. This study aimed to develop a tool to quantify household cooking environments and establish its content validity, face validity, and inter-rater agreement. Methods: The Home Cooking EnviRonment and equipment Inventory observation form (Home-CookERI™) was developed as a 24-question (91-item) online survey. Items included domestic spaces and resources for storage, disposal, preparation, and cooking of food or non-alcoholic beverages. Home-CookERI^TM^ was piloted to assess content validity, face validity, and usability with six Australian experts (i.e., dietitians, nutrition researchers, chefs, a food technology teacher, and a kitchen designer) and 13 laypersons. Pilot participants provided feedback in a 10 min telephone interview. Home-CookERI™ was modified to an 89-item survey in line with the pilot findings. Inter-rater agreement was examined between two trained raters in 33 unique Australian households. Raters were required to observe each item before recording a response. Home occupants were instructed to only assist with locating items if asked. Raters were blinded to each other’s responses. Inter-rater agreement was calculated by Cohen’s Kappa coefficient (κ) for each item. To optimize κ, similar items were grouped together reducing the number of items to 81. Results: Home-CookERI^TM^ had excellent content and face validity with responding participants; all 24 questions were both clear and relevant (X^2^ (1, *n* = 19; 19.0, *p* = 0.392)). Inter-rater agreement for the modified 81-item Home-CookERI™ was almost-perfect to perfect for 46% of kitchen items (*n* = 37 items, *κ* = 0.81–1), moderate to substantial for 28% (*n* = 23, *κ* = 0.51–0.8), slight to fair for 15% (*n* = 12, *κ* = 0.01–0.5), and chance or worse for 11% of items (*n* = 9, *κ* ≤ 0.0). Home-CookERI^TM^ was further optimized by reduction to a 77-item version, which is now available to researchers. Conclusion: Home-CookERI™ is a comprehensive tool for quantifying Australian household cooking environments. It has excellent face and content validity and moderate to perfect inter-rater agreement for almost three-quarters of included kitchen items. To expand Home-CookERI™ applications, a home occupant self-completion version is planned for validation.

## 1. Introduction

Food skills, cooking skills, and home cooking frequency are associated with higher diet quality [1,2,3,4,5,6] and improved health outcomes, including lower risk for obesity [5,7,8] and type 2 diabetes [8]. However, since the 1960s, involvement in food preparation and time spent cooking at home has decreased in many western countries [9,10,11]. Scientific understanding of what “home food preparation” or “home cooking” behaviors encompass, and how to consistently measure these behaviors is still developing [4,9,10,11,12,13,14,15]. Informed by the social ecological model [16], home cooking behavior can be influenced by a myriad of individual, interpersonal, and environmental factors which include but are not limited to food and cooking skills, values and enjoyment of cooking, education and nutrition knowledge, culture and ethnicity, and resources (i.e., time, money, cooking spaces, and equipment) [4]. To better understand the association between cooking, dietary intake, and health, robust scientific approaches and measurement tools specific to home cooking are required. High quality validated measures to assess food and cooking skills are available, [17] but in order to gain a more comprehensive understanding of home cooking behavior, these existing tools must be further complemented by specific instruments for assessing more facets of the socioecological influences on home cooking behavior.

One aspect that is beginning to be explored is the Home Cooking EnviRonment, which is defined as the spaces and equipment available for domestic food preparation and cooking, and its impact on diet quality. Exploration of the international literature has identified six questionnaires from seven papers that measure the Home Cooking EnviRonment in the United States of America (US) and France [6,18,19,20,21,22,23]. All tools vary markedly in length, containing between 4 and 44 different kitchen items, with most lacking detail of scientific rigor during development and varying in scientific quality. None of the tools identified could be considered a gold standard for measuring the Home Cooking EnviRonment and equipment.

One of the US tools assessed housing quality and requirements in low-income Latino farmworker families [18]. The tool was created from the housing quality standards promulgated by the US Department of Housing and Urban Development for the Housing Choice Voucher Program without any validation procedure reported.

Another US study used a different tool to investigate the availability of 19 different food preparation items, represented with pictures and text, to tailor nutrition education programs for low income clients applying for a Food Stamp Program [19]. The authors identified methodological shortcomings of the tool such as the absence of any validation procedures, the high likelihood of missing some basic kitchen equipment and not assessing functionality of electric appliances. They also highlighted a need to confirm construct validity of the tool.

Two other US studies [6,23] used a 41-item Food Preparation Checklist, developed specifically for those studies in consultation with a registered dietician and without scientific validation, to examine the association between household food preparation supplies and dietary behaviors in 6–13-year olds.

Another US intervention study in low income families created a 35-item inventory of basic cooking utensils (e.g., measuring cups), tableware and eating utensils (e.g., plates, silverware), cookware (e.g., pan, skillet), appliances (e.g., working oven), and other resources (e.g., table and chairs) to predict children’s adiposity indices [21]. Details regarding the development and validity testing of the inventory were not reported.

Two other tools—the comprehensive home environment survey (CHES) [20] from the US and a set of questions about home cooking resources in a supplemental component of the NutriNet-Santé Study [22] from France—had a more rigorous development process but had other limitations. The 181-item CHES had a structured development procedure, including face, content, predictive, and concurrent validity, as well as inter-rater agreement between two caregivers and test–retest agreement, but the comprehensiveness of the kitchen-subscale was limited to only seven items [20]. The tool used in the cross-sectional NutriNet-Santé Study was a web-based, self-reported questionnaire that was assessed for face and content validity. The food preparation behaviors section was also limited to seven utensils and appliances (pressure cooker, zester, baking pan, measuring cup, food processor, gas oven, or electric oven). The item list was derived from French statistics on income and living conditions. Among volunteers from the general adult population (n = 60,000) who completed the questionnaire, food preparation behaviors were not significantly different across socioeconomic groups. However, a larger range of kitchen equipment was observed in households with high income [22]. While both tools [20,22] were assessed for face and content validity, neither have been assessed for concurrent validity against a gold standard and both are relatively short kitchen sub-scales which limits their capacity to comprehensively measure Home Cooking EnviRonments and equipment [20,22].

In order to robustly test hypotheses regarding the impact of the Home Cooking EnviRonment and equipment on diet quality and/or health in future studies, a ‘gold standard’ survey tool to measure Home Cooking EnviRonments and equipment is needed. It is proposed that such a tool can be developed by applying a standard scientific framework [24,25] to establish face and content validation, and inter-rater agreement of the survey tool where the raters are content experts and independently perform direct observations of the same Home Cooking EnviRonments.

This study aimed to develop a gold standard tool, the Home Cooking EnviRonment and equipment Inventory observation form (Home-CookERI™) to identify the presence of spaces, surfaces, and equipment (e.g., heavy and small kitchen appliances, kitchenware, cleaning facilities, access to recipes) in Australian households that are used for the storage, preparation, or cooking of food or non-alcoholic beverages in the kitchen and elsewhere in a residence.

## 2. Materials and Methods

### 2.1. Study Design

A four-stage-mixed-method approach was used to develop Home-CookERI^TM^ using qualitative and quantitative measures with differing population samples, as summarized in Table 1, to establish the gold standard measure. First, our research team developed the initial draft of the online Home-CookERI^TM^ tool (Stage 1). Next qualitative procedures were used to establish face and content validation of Home-CookERI^TM^ with experts and laypersons to ensure its clarity, relevance, comprehensibility, and user-friendliness [25,26,27] (Stage 2). Subsequently, visits were made to households to assess inter-rater agreement of the Home-CookERI^TM^ tool when completed by trained members of the research team [25,27,28,29,30] (Stage 3). Ultimately, the final version of Home-CookERI^TM^ was produced including all adaptations from the previous stages (Stage 4).

Participants were recruited via networks of the research team using e-mail invitations and from previous research team contacts who had agreed to be re-contacted for future research participation.

The study was approved by the Human Research Ethics Committee of The University of Newcastle (UON), Australia, Approval No. H-2019-0189. Participants in Stages 2 and 3 provided informed consent to participate via an online form in Qualtrics^®^.

Each development stage is described in more detail in Table 1 and Table 2 and in Appendix B.

#### 2.1.1. Stage 1: Home-CookERI^TM^ Initial Design

The initial version of Home-CookERI^TM^ was developed iteratively by the authors, i.e., eight of whom have nutrition and cooking expertise with one or more of the following qualifications/roles: dietitian (*n* = 5), chef (*n* = 1), nutrition researcher (*n* = 7), and home economist (*n* = 1). Statistical considerations for the development of Home-CookERI^TM^ was guided by two authors (i.e., C.O., M.F.).

The initial survey included 24 questions covering 91 items likely to be located in contemporary Australian Home Cooking EnviRonments (see Table 2). Home-CookERI™ was developed as an online survey using Qualtrics^®^ allowing for completion on smart phones, tablets, or computers. Most questions grouped similar items (e.g., manual tools for food preparation tasks: slotted spoon, ladle, whisk, spatula) and each question was displayed on a single page to support logical completion. Home-CookERI^TM^ was divided into eight sections, as described in Table 2. The naming and definition of each section in the survey was determined within the research team. All questions were multiple choice items with an option to declare that the participant did not have the listed items. Questions were in text format, with pictures only used to assist in distinguishing different types of knives. Fridges could be distinguished by sizes, chopping boards by quantity and questions about pots, pans, frypans, and mixing bowls provided options to distinguish both quantity and size. The initial selection of items was based on team members’ observations of Australian kitchenware websites, department stores, cookbooks, personal recall of items previously observed in Australian Home Cooking EnviRonments, contents of The University of Newcastle teaching food laboratories, and reviewing existing kitchen checklists used by international researchers [6,18,19,20,21,22]. Inclusion criteria for kitchen items were to adequately represent the construct of interest, i.e., spaces, surfaces, and equipment in Australian households used for the storage, preparation, or cooking of food or non-alcoholic beverages in the kitchen and elsewhere in a residence. The authors conducted a qualitative pretest, where each survey item was discussed step-by-step to identify divergence based on deep knowledge of the research aims [31]. The initial 91 included items were thought to be an optimum number to allow collection of comprehensive data about the Home Cooking EnviRonment and equipment while minimizing response burden [32].

#### 2.1.2. Stage 2: Face and Content Validity and Usability Testing

Face and content validity were assessed from August-September 2019. Seven authors and six nutrition and cooking experts from a convenience sample (i.e., dietitians, nutritionists, researchers in nutrition-related fields, chefs/cooks, home economists/food technology teachers, cooking program facilitators, and experts on kitchen architecture/cooking equipment design) qualitatively assessed the face and content validity of Home-CookERI^TM^ via a Qualtrics^®^ survey. Other inclusion criteria for participants were as follows: age 18 years or older, access to the internet, and provision of informed consent to participate. Nutrition and cooking experts reviewed all 24 questions and 8 content definitions of the sections (Table 2) in Home-CookERI^TM^, rating each question/section definition for clarity (conciseness, use of correct grammar and presentation in an appropriate format) and relevance to domestic cooking-environments. Further space was provided for the expert participants to provide additional comments on each question or to suggest specific revisions [33]. Further information on the survey review is provided in Appendix A and Appendix B. Experts were invited to participate in a subsequent “respondent debriefing”, i.e., short face-to-face or phone interview [30,32,34], lasting up to 15 min, with a member of the research team (SS). They were asked to ‘think aloud’ and to further discuss their ratings and additional comments about survey clarity, meaningfulness, purposefulness and completeness [25,28,29].

Home-CookERI^TM^ was further assessed for face and content validity by a convenience sample of 13 “laypersons” (i.e., not having any professional background in the field of the “nutrition and cooking experts”) from the general adult population. The other inclusion criteria and assessment procedures were the same as those for the expert group, except all participants needed to live in Australia.

Finally, based on the feedback from experts and laypersons, a revised version of Home-CookERI^TM^ was created for Stage 3, containing 38 questions and covering 89 items of the Home Cooking EnviRonment (see Table 2).

#### 2.1.3. Stage 3: Inter-Rater Agreement

In September–October 2019 the revised version of Home-CookERI^TM^ was used to assess inter-rater agreement of the tool in a home visit study involving volunteer participants. Participants were a convenience sample of adults from 33 different households in Newcastle, Australia. This sample size was chosen prospectively as a compromise between practical limitations and operating characteristics, and would enable estimation of a Kappa statistic with 95% confidence interval having a 25% margin of error, assuming a true Kappa of 0.75 (estimated using the kappa size package in R) [35,36]. Raters who performed the home visits were dietician researchers (S.S., R.A., V.S.) and a student dietician (S.St.) from UON. All raters attended a 2-h training session where they were briefed about home visit safety and the inter-rater agreement research protocol. The UON health and safety policies and procedures were adhered to throughout. In order to test the research protocol, raters performed an assessment with Home-CookERI^TM^ in a training kitchen at UON.

Two raters were selected for each visit based on their availability at the time that was nominated by the participant. All combinations of raters were used for the 33 home visits. The two raters independently completed Home-CookERI^TM^ in each participant’s home. The first rater completed Home-CookERI^TM^, while the second rater completed an audio activity on an electronic device wearing headphones and, when possible, situated themselves in an adjacent room. This protocol was used so that the second rater was audio and visually blinded to the first rater’s completion of Home-CookERI^TM^. Then the second rater completed Home-CookERI^TM^. Participants (i.e., the home occupant) were instructed to assist with locating items only if asked by the rater to prevent them from biasing a rater’s completion of Home-CookERI^TM^. To allow rater’s responses for a specific participant’s household to be linked for analysis, each household was allocated an identification number. Raters were also provided with an individual identification number to enter on each completion of Home-CookERI^TM^.

#### 2.1.4. Stage 4: Optimization of Home-CookERI^TM^

Home-CookERI^TM^ was revised again after the inter-rater agreement evaluation, where kitchen items were specified, generalized, collapsed, or removed according to the findings, to increase the validity and inter-rater agreement of the tool. For more details please refer to Table 2 and Table 3 and Appendix B. The final tool is available for researchers to use (see Appendix C) and can be shared as a Qualtrics^®^ survey upon request and approval of the corresponding author.

### 2.2. Data Analysis

For Stage 2 descriptive statistics were calculated in Excel for the initial version of Home-CookERI^TM^. A two-tailed t-test was carried out to determine consistency between experts and laypersons on their judgement of clarity and relevance of the 24 questions and 8 content definitions of the sections. In Stage 3, all 89 Home-CookERI^TM^ kitchen items produced categorical, binary data which were analyzed for inter-rater agreement using Cohen’s Kappa coefficients. These were calculated using Stata statistical software (StataCorp, 2017, Stata Statistical Software: Release 15. College Station, TX: StataCorp LLC.).

Kappa values (*κ*) can range from −1.0 (agreement worse than expected, or systematic disagreement) to 1.0 (perfect agreement between raters). Random chance agreement is represented by “0”. Due to diverging recommendations about reasonable Kappa levels [25,28,30,37], a subject-sensitive approach to define levels of inter-rater agreement across multiple data collectors was carried out where “almost-perfect” to “perfect” inter-rater agreement was defined as *κ* = 0.81 to *κ* = 1.0, respectively. “Moderate” to “substantial” inter-rater agreement for the identified home cooking items was determined as *κ* = 0.51 to 0.8, “slight” to “fair” inter-rater agreement as *κ* = 0.01 to 0.5 and “no” inter-rater agreement when items presented as *κ* ≤ 0.0.

## 3. Results

### 3.1. Results Stage 2: Content and Face Validity

The pilot test found Home-CookERI^TM^ had excellent content and face validity with participants responding that all 24 questions were both clear and relevant (X^2^ (1, *n* = 19; 19.0, *p* = 0.392)). No significant differences were found within the respective ratings for clarity and relevance from experts and laypersons. Findings from Stage 2 helped to determine which items needed reviewing or deletion and where ungrouping of some items into single questions and changing the question order were necessary to optimize the tool. More details of changes can be retraced in Table 2 and Appendix B. Other key adjustments included formatting Home-CookERI^TM^ for ease of use during home visits in Stage 3 of development, such as displaying all questions on one page to allow raters to easily navigate the tool if items were initially missed and viewed later during an observation. At the end of each section a space was provided for the rater to record notes during the inter-rater agreement procedure (Stage 3). Following these changes, the total assessable kitchen items were reduced to 89 at the commencement of Stage 3 (Table 2 and Appendix B).

### 3.2. Results Stage 3: Inter-Rater Agreement

Table 3 reports the Kappa values and inter-rater agreement across 81 assessable kitchen items at the end of Stage 3, which originally were 89 items at the beginning of Stage 3. This reduction was performed to optimize the tool’s inter-rater agreement. Specifically, seven items were collapsed to improve inter-rater agreement (e.g., different sizes of pots/saucepans, pans, mixing bowls, fridges without freezer) and one item was removed (“other storage spaces in the kitchen”). The latter item was originally included to ascertain if all kitchen storage spaces had been identified. All answering options “other–please specify” and “I have none of these items” are excluded from Table 3, leading to Kappa values for a total of 81 kitchen items. For those 81 assessable kitchen items almost perfect to perfect agreement between raters was found in 46% of the items (*n* = 37, raters = 2, *κ* = 0.81–1). However, for 7 of the 23 observable kitchen items scoring perfect agreement (see Table 3), no variance was found amongst rater responses (100% “yes” or 100% “no”) that yielded an incalculable Kappa value. Inter-rater agreement was moderate to substantial for 28% of items (*n* = 23, raters = 2, *κ* = 0.51–0.8) and slight to fair for 15% of items (*n* = 12, raters = 2, *κ* = 0.01–0.5). Agreement between raters could not be determined in 11% of the items (*n* = 9, raters = 2, *κ* ≤ 0.0) and is explained further in the discussion.

### 3.3. Stage 4

Based on the findings (Stage 3), 12 items were removed or combined to produce the final Home-CookERI^TM^ with 77 items (Appendix C). Due to poor inter-rater agreement for the paring knife (*κ* = 0.35), it was collapsed into one item with the utility knife. Kitchenware for storage of food and beverages was collapsed with kitchenware for transportation of food and beverages “to go”. Other items where inter-rater agreement could not be determined were removed: “Any storage space indoors and outside kitchen” (*κ* = −0.10) and “electric pressure cooker” (*κ* = −0.03). The final Home-CookERI^TM^ tool does not include non-assessable answering options such as “Other”.

## 4. Discussion

Home-CookERI^TM^ has high face and content validity (Stage 2), indicating the relevance of its contents based on lay and expert participants’ subjective judgements. Stage 2 participants reported Home-CookERI^TM^ questions as being clear and meaningful for the purpose of measuring Home Cooking EnviRonments. Inter-rater agreement for the majority of the assessable Home-CookERI^TM^ items was moderate to perfect.

Findings confirm Home-CookERI^TM^ is a suitable tool to identify the presence of spaces, surfaces, and equipment in households for food and non-alcoholic beverage storage, preparation or cooking. Cohen’s Kappa levels of κ ≥ 0.81 indicate good agreement in relevant scientific literature [38,39]. From a topic-specific point of view, in this study a more modest agreement coefficient for acceptable inter-rater agreement levels was set. In 74% of the cases, raters identified the kitchen items with a Kappa of κ ≥ 0.51, indicating “moderate” to “perfect” agreement. As Home-CookERI^TM^ assesses the presence of items for the purpose of the preparation of food and non-alcoholic beverages, this result is considered to be very good.

The best inter-rater agreement results were found for items that had high levels of agreement for both “yes (present)” and “no (absent)” and that allowed for higher discrimination as Kappa requires both agreement outcomes to calculate a value [38,39].

“Pseudo-perfect” agreements happened for broadly available items in this study. For example, cold/hot water tap and stovetop were present in all assessed homes and identified as such by all raters (100 % inter-rater agreement; *κ* = 1.0). However, the presence of these items was a reflection of the few assessed households rather than “true” perfect inter-rater agreement for both “yes” and “no” responses. By increasing the sample size to include more socio-economically diverse households, a better distribution of both “yes” and “no” responses could have been reached, which would have produced a balanced Kappa equation. Other “false perfect” agreements were found for example for the “electric food processor/blender of any kind” (*κ* = 1), where the description of the item was not specific enough to produce variance, i.e., it only produced “yes” answers. This machine was interpreted as anything from a stick blender, a hand-mixer or benchtop food processor and is similar to the “mixer question”. It was kept in the final tool however to account for the presence of any electric appliance which would fulfill the function of blending/mixing.

Limited response variability impacted the results even further, where a single discrepant “yes” and “no” amongst otherwise perfect agreement in all other answers occurred. These “rare cases” produced a heavily weighed Kappa denominator which skewed the value negatively and resulted in *κ* ≤ 0, despite the raw inter-rater agreement ratio being 99.8%. This coincidence is a well-known limitation of Cohen’s Kappa [39].

For Kappa values *κ* < 0.5, it was assumed that rater agreements were more likely to occur by chance, such as for the differentiation of fridges and freezers. When analyzing potential root causes for uncertainties, potential sources of confusion were identified, for example the differentiation between the definition and the functionality of kitchen items. This also happened for the slotted spoon (*κ* = 0.45). It was discovered that one rater confirmed its presence when an object looked similar to the way it would be illustrated in department store websites or in cookbooks, whereas another rater confirmed its presence if any type of utensil was found in the kitchen which could fulfill similar functions as the slotted spoon.

Discriminant sizing of similar items led to poorer inter-rater agreement across specific Home-CookERI^TM^ items (for example small and large saucepans). To create a more harmonious research tool it was agreed to condense such items, see Appendix B.

In contrast to the above-mentioned skewed Kappa values due to rare cases, “true” negative Kappa values are an important issue for inter-rater agreement, as this usually indicates a systematic disagreement among raters, and a significant design flaw. In line with literature recommendations, the interpretability of the respective items in Home-CookERI^TM^ and the understanding of the raters were analyzed [38,39]. Kappa values below zero were found for five “unclear items” in the list of 81 assessable items in Home-CookERI^TM^ at Stage 3. They were either not observable and therefore reliant on the answer of the participants, e.g., “any space…” (*κ* = −0.10) where food is stored or prepared or designations were confusing, e.g., “baking tray” (*κ* = −0.04) versus “cookie sheet” (*κ* = 0.14). Items with a negative κ were removed, adapted, or collapsed (Appendix B).

These examples demonstrate the complex challenge of the unambiguous identification of items related to Home Cooking EnviRonments, even despite direct observations from experts. Differences in culture, socioeconomic background, and cooking skills are just three of many aspects that influence if and how items are recognized. To address this, the use of pictures for each item was considered, but was decided that this may limit rater’s interpretation of the variation in appearance of a particular item. However, the use of pictures was tested for the “knives-question” in Home-CookERI^TM^ and could confirm good inter-rater agreement for three of the five knife types. The utility and paring knives were regularly used interchangeably, likely due to their similarity in appearance and were therefore collapsed for the final Home-CookERI^TM^ tool.

Previous studies either did not perform any structured validation procedures [6,18,19,21] or only consisted of a very small number of items [20,22]. One of the most important strengths of Home-CookERI^TM^ is that it does not rely on subjective means of assessment as currently available questionnaires do. Home-CookERI^TM^ is not only the first scientifically evaluated tool to comprehensively measure items for home cooking but it has novelty character being an objective gold standard measure to quantitatively assess domestic kitchens and which can be used by anyone with a background in nutrition research or people who are trained by this group. Findings confirm Home-CookERI^TM^ to be easy to understand and scientifically robust for identifying the presence of spaces, surfaces and equipment in households for food and non-alcoholic beverage storage, preparation, or cooking. Additionally, the final survey consisting of 77 items has low response burden with a predicted duration of 12 min for completion according to the “Expert Review” function from the Qualtrics^®^ software.

Limitations of this study include the use of convenience samples from the general population. Study participants came from the researchers’ networks and that may have influenced face and content validity results for Home-CookERI^TM^. Similarly, the convenience sample of experts chosen for the Home-CookERI^TM^ development might have impacted the agreement about the relevance of certain items. On the other hand, as Home-CookERI^TM^ is aimed at researchers in the field of nutrition, the constituted sample of dietitians, nutritionists, researchers in nutrition-related fields, chefs/cooks, home economists/food technology teachers, cooking program facilitators, and experts for kitchen architecture/cooking equipment design, can be considered a strength to judge items in the tool as they are the ones who are most likely using it in their professional work. As data saturation about the consensus of clarity and relevance of the items among all participants was achieved, the sample was further confirmed as purposeful and the relevance of their judgement unambiguous [40]. Blinding was used between raters when assessing home environments. This strengthens the validity of findings within the study via elimination of potential observer biases.

The development of the Home-CookERI^TM^ tool as a gold standard, through investigation of actual Home Cooking EnviRonments by trained raters, was the first logical and most essential step to establish a robust assessment tool in this field. This study provides a methodology for researchers in other countries to replicate to create a locally relevant gold standard tool.

Future research should investigate validating the tool for self-completion by laypersons as this would allow it to be used with confidence in population-based studies. Such a tool would enable the conduct of research to facilitate a greater understanding of the link between Home Cooking EnviRonments, cooking behaviors (including cooking/food skills and confidence), eating patterns, diet quality and health in people with diverse socioeconomic backgrounds.

A self-completion version of Home-CookERI^TM^ could also be used in clinical and public health settings to enable the personalization of recipe provision and cooking skill education to individual’s home cooking resources. Pictures could then be considered to clarify the items in question and to better suit specific groups of end-users. Development of context-specific versions of Home-CookERI^TM^ may be beneficial. Whilst for the design of a cooking class it might be reasonable to ask whether people have particular tools available to perform preparation tasks for specific dishes, epidemiological research could aim to build rankings of kitchen equipment quality and potentially include frequency of use. While a poor cooking environment might be expected to limit the ability to prepare healthy diets, some households might harbor tools which are never used. A cooking resource usage score could help distinguish between the presence of the items, versus their frequency of use when adding a weighted analytic component. Looking at correlations between resource usage, cooking skills, and diet quality could be a highly relevant field for future research.

## 5. Conclusions

Presently Home-CookERI^TM^ is the only domestic kitchen inventory tool with high-level face and content validity and inter-rater agreement confirmed via direct observation of Home Cooking EnviRonments. Home-CookERI^TM^ can be recommended for use in studies when Australian Home Cooking EnviRonments are observed by researchers or it could be adapted to reflect other eating cultures. Future research is needed to adapt the Home-CookERI^TM^ tool for self-completion by the general population.

## Figures and Tables

**Table 1 nutrients-12-01853-t001:** Home Cooking EnviRonment and equipment Inventory observation form (Home-CookERI™) four-stage development process.

Stage	Name of Stage	N
1	Home-CookERI^TM^ initial design	All authors
2	Face and content validity, usability testing	19 ^1^
3	Inter-rater agreement	33
4	Optimization of Home-CookERI^TM^	n.a.

^1^ Until data saturation achieved. Included 6 experts and 13 laypersons in addition to the authors.

**Table 2 nutrients-12-01853-t002:** Home-CookERI^TM^ Structure, Stages 2–4.

Sections and Content Definitions	Number of Questions Stage 2	Number of Kitchen Items Stage 2	Number of Questions Stage 3	Number of Kitchen Items Stage 3	Number of Questions Stage 4	Number of Kitchen Items Stage 4
Spaces/surfaces: food storage, preparation, cleaning, disposal	3	13	3	14	3	12
Heavy kitchen appliances (large machines for routine housekeeping tasks around cooking, washing, food preservation, used with electricity or fuel)	5	11	5	10	5	9
Small kitchen appliances: food preparation/cooking with heat (portable/semi-portable, often placed on benches/other household-platforms, in or outside the house, used with electricity or fuel)	3	17	5	14	7	15
Small kitchen appliances: mixing, processing, blending, juicing (portable/semi-portable, often placed on benches/other household-platforms, in or outside the house, used with electricity or fuel)	1	3	5	5	3	3
Kitchenware: food preparation/cooking (cookware to prepare food and drinks on a stove/cooktop)	3	10	6	10	5	5
Utensils: food preparation/cooking (small tools, used for different tasks in the food preparation process, e.g., cutting, grating, mixing, stirring, turning, frothing, storing and measuring)	7	26	10	26	10	23
Kitchenware: baking (cooking vessels intended for use inside an oven)	1	8	1	7	2	8
Food/drink storage containers	1	3	3	3	2	2
**Total**	**24**	**91**	**38**	**89**	**37**	**77**

**Table 3 nutrients-12-01853-t003:** Home-CookERI^TM^ Kappa values for inter-rater agreement at the end of Stage 3.

Items (*n* = 81)	Kappa Value *κ*	Inter-Rater Agreement Ratio (n%)
Kitchenette ^a^	1.00	100.00
Sink ^a^	1.00	100.00
Cold water tap ^a^	1.00	100.00
Hot water tap ^a^	1.00	100.00
Chest freezer (large box or chest with a hinged lid) ^a^	1.00	100.00
Fixed Stovetop/Cooktop ^a^	1.00	100.00
Oven that is fixed in place	1.00	100.00
Air fryer	1.00	100.00
Electric grill press/Sandwich maker	1.00	100.00
Electric food processor/Blender of any kind ^a^	1.00	100.00
Electric kettle	1.00	100.00
Saucepans/Pots of different sizes: small, medium, large ^b^	1.00	100.00
Pans (Frypan/Sauté pan/Grill pan/Wok): small, medium, large ^b^	1.00	100.00
Whisk	1.00	100.00
Rolling pin	1.00	100.00
Mixing bowls of different sizes: small, medium, large ^b^	1.00	100.00
Measuring jugs	1.00	100.00
Colander/Strainer	1.00	100.00
Cook’s/Chef’s knife	1.00	100.00
Kitchen shears/Scissors	1.00	100.00
Manual peeler	1.00	100.00
Bread/Loaf pan	1.00	100.00
Food canisters for storage	1.00	100.00
Electric coffee machine	0.93	96.97
Electric Frypan/Wok	0.93	96.97
Electric All-in-one appliance (chops, mixes, cooks)	0.92	96.97
Rangehood/Exhaust fan	0.91	96.97
Dishwasher	0.91	96.97
Chopping boards	0.91	96.97
Measuring cups	0.89	96.97
Cake pan	0.89	96.97
Steamer basket/Insert for saucepans	0.87	96.97
Salad spinner	0.87	93.94
Cooling rack	0.86	93.94
Fridge, without freezer (any size) ^b^	0.84	96.97
Microwave	0.84	96.97
Electric slow cooker	0.82	90.91
Measuring spoons	0.79	90.91
BBQ	0.79	90.91
Electric multi cooker	0.76	93.94
Ladle	0.76	93.94
Electric rice cooker	0.74	90.91
Recycling bin/Compost bin	0.72	93.94
Garage	0.72	93.94
Potato masher	0.71	93.94
Outdoor kitchen/BBQ area	0.69	84.85
Deep fryer	0.65	96.97
Toaster	0.65	96.97
Manual grater	0.65	96.97
Drinking bottles/Mugs to go	0.65	96.97
Manual tin/can opener	0.64	93.94
Fridge, with freezer (any size)	0.64	93.90
Scales	0.63	84.85
Dutch oven/Casserole	0.62	81.82
Electric mixer of any kind	0.61	84.85
Electric juicer	0.60	87.88
Carving knife	0.55	78.79
Stove top pressure cooker	0.52	90.91
Serrated bread knife	0.52	90.91
Pantry/Cupboards/Other storage spaces NOT in the kitchen	0.52	78.79
Portable fridge or freezer	0.48	93.94
Standalone upright freezer (several compartments)	0.46	81.82
Slotted spoon	0.46	84.85
Pie tin	0.38	69.70
Roasting pan	0.37	90.91
Utility knife	0.37	90.91
Paring knife	0.35	90.91
Walk in pantry	0.22	84.85
Electric steamer	0.20	84.85
Benchtop oven	0.20	84.85
Deep dish	0.19	75.76
Cookie sheet	0.14	57.58
Cellar	0.00	96.97
Rubbish bin/Garbage disposal	0.00	93.94
Utensils to turn hot food, e.g., Spoon/Turner/Spatula/Tong	0.00	96.97
Cookbook/Internet	0.00	96.97
Electric pressure cooker	−0.03	93.94
Food canister to go	−0.03	93.94
Pantry/Cupboard in kitchen	−0.04	90.91
Oven/Baking tray	−0.04	90.91
Any storage space indoor, but outside kitchen	−0.10	75.76

^a^ “False perfect” inter-rater agreement due to missing variance, i.e.,100% “yes” or 100% “no”. ^b^ Collapsed to compute inter-rater agreement statistics in Stage 3.

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
