# Peer review of "Development of the Home Cooking EnviRonment and Equipment Inventory Observation form (Home-CookERITM): An Assessment of Content Validity, Face Validity, and Inter-Rater Agreement"

_nutrients, 2020, doi:10.3390/nu12061853_

Round 1

Reviewer 1 Report

This paper presents the validity testing of a questionnaire developed to quantify household cooking environments. The final survey offers a tool which can be used in various applications including public health nutrition, nutritional epidemiology, health promotion, and evaluation of nutrition interventions. The survey would be suitable for international use, although some minor changes may need to be made to ensure terminology of cooking equipment is appropriate for the population it is being used with. The paper is well written and clearly states the validation process and amendments that were made to the survey throughout the validation process.

Few minor queries / suggestions:

In the title of the paper and in the methods and introduction – why do observation and from have capital letters but equipment doesn’t. I realise that equipment does not form part of CookERI but neither do observation and form.

Lines 86 and 87 – commas not needed after e.g.

Line 99 – n=60,000

End of introduction and on occasion in the other sections – paper slips into first person – is this in line with the journal style?

Line 120 and on all occasions where index markers are placed – showing as errors, this makes it difficult to know what table / figure is being referred to and interferes with the sense of line 248 and line 260

Many hyphens could be removed (particularly after face / content) including those on lines 122, 136 (stage 2), 166, 167 and 169, Table 2 (2 instances of in-), lines 241, 242 and 285, Appendix B (Stage 1 and Stage 2)

Line 131 is the Ethics number written correctly as there is an H- outside and within the square brackets

Many uses of z instead of s – is the journal style American or UK English? E.g. optimization in table 1 and other places throughout the manuscript

Line 142 – unclear of what the sentence “Two authors ensured statistical quality (CO, MF)” relates to - if it is meant to the be there CO and MF should be C.O. and M.F. in line with the rest of the manuscript.

Line 152 – think “which” needs to be inserted between “pictures” and “were”

Table 2 – Left justify column 1

Line 211 – why is “(i.e. homeowners)” needed – this is the first mention that the participants were such and is it necessary to know this – did all participants indeed own their homes?

Line 221 – “be used” reads better changed to “use”

Line 235 – change the two “ands” to “to” to be consistent with the rest of the sentence

Line 253 – should “in” be changed to “for” as you are reporting what is about to happen

Table 3 – consistency with capitals in column 1

Lines 273-283 – why is this a series of one sentence paragraphs?

Line 352 – “nutrition” instead of “nutritional”?

Appendix A and Figure 1 titles are the same - why have both?

Appendix B and Figure 2 titles are the same – why have both?

References 34 and 39 appear incomplete and is author correct for reference 35?

Author Response

Author's Reply to the Review Report (Reviewer 1) Reviewer comment

Response

Line number after revision

This paper presents the validity testing of a questionnaire developed to quantify household cooking environments. The final survey offers a tool which can be used in various applications including public health nutrition, nutritional epidemiology, health promotion, and evaluation of nutrition interventions. The survey would be suitable for international use, although some minor changes may need to be made to ensure terminology of cooking equipment is appropriate for the population it is being used with. The paper is well written and clearly states the validation process and amendments that were made to the survey throughout the validation process.

Thank you very much for your positive feedback.

n/a

In the title of the paper and in the methods and introduction – why do observation and from have capital letters but equipment doesn’t. I realise that equipment does not form part of CookERI but neither do observation and form.

Throughout we have removed capital letters from observation and form to address this inconsistency.

n/a

Lines 86 and 87 – commas not needed after e.g.

We have omitted the commas.

n/a

Line 99 – n=60,000

Thank you, we added “=”

97

End of introduction and on occasion in the other sections – paper slips into first person – is this in line with the journal style?

Thank you for identifying this inconsistency. We have searched for use of first person, “we” and “our”, and updated these sentences.

n/a

Line 120 and on all occasions where index markers are placed – showing as errors, this makes it difficult to know what table / figure is being referred to and interferes with the sense of line 248 and line 260

We used automized cross-referencing for all tables and figures in the manuscript. These must have been removed during the editing process. We checked and manually repasted them again.

117, 241, 258 and throughout

Many hyphens could be removed (particularly after face / content) including those on lines 122, 136 (stage 2), 166, 167 and 169, Table 2 (2 instances of in-), lines

We have removed all hyphens in “face / content validity” expressions.

Various

Reviewer 2 Report

Please see the paper. I inserted my suggestions using the comment tab in adobe.

Line 38: This is confusing. You state 91 and 81 questions in the methods.

Line 91: What were these limitations? You talk about the limitations in the next two paragraphs. I suggest that you combine the subsequent two paragraphs into this paragraph.

Line 120: Please add a reference.

Line 132: Remove this parathesis.

Line 324: Is it important for cooking to have a "slotted" spoon or something that functions as a slotted spoon?

Author Response:

Line 38: This is confusing. You state 91 and 81 questions in the methods.

Thank you for this remark. This is indeed a linguistically tricky part. We have updated all relevant sections of the manuscript to make this aspect clearer. (Line number after revisions: 38)

Line 39: Please check--a phrase is repeated.

We have revised the text to make this sentence clearer. ( Line number after revisions: 37-42)

Line 91: What were these limitations? You talk about the limitations in the next two paragraphs. I suggest that you combine the subsequent two paragraphs into this paragraph.

Very good point. We combined the paragraphs so that discussion of the limitations of references 20 and 21 is within one paragraph. ( Line number after revisions: 90)

Line 120: Please add a reference.

We used automized cross-referencing for all tables and figures in the manuscript. These must have been removed during the editing process. We checked and manually repasted them again.

Line 132: Remove this parathesis.

Thank you. We removed the parenthesis. (Line number after revisions: 129)

Line 324: Is it important for cooking to have a "slotted" spoon or something that functions as a slotted spoon?

Our aim was to create a comprehensive tool with common items in Australian kitchens. Your question is one we have discussed at length during the development of Home-CookERI. We attempted to consistently identify equipment based on its commonly recognized name rather than function as we felt this was more objective in any context compared with an assessment of function in one home environment. At this point of the discussion we explain potential root causes for inter-rater agreements which occurred by chance only. The slotted spoon serves as an example for other kitchen items, where confusion about the differentiation between the definition and the functionality occurred and which led to diverging inter-rater agreement. (Line 312 ff)